# Modifier Genes in Microcephaly: A Report on *WDR62*, *CEP63*, *RAD50* and *PCNT* Variants Exacerbating Disease Caused by Biallelic Mutations of *ASPM* and *CENPJ*

**DOI:** 10.3390/genes12050731

**Published:** 2021-05-13

**Authors:** Ehtisham Ul Haq Makhdoom, Syeda Seema Waseem, Maria Iqbal, Uzma Abdullah, Ghulam Hussain, Maria Asif, Birgit Budde, Wolfgang Höhne, Sigrid Tinschert, Saadia Maryam Saadi, Hammad Yousaf, Zafar Ali, Ambrin Fatima, Emrah Kaygusuz, Ayaz Khan, Muhammad Jameel, Sheraz Khan, Muhammad Tariq, Iram Anjum, Janine Altmüller, Holger Thiele, Stefan Höning, Shahid Mahmood Baig, Peter Nürnberg, Muhammad Sajid Hussain

**Affiliations:** 1Cologne Center for Genomics (CCG), Faculty of Medicine and University Hospital Cologne, University of Cologne, 50931 Cologne, Germany; shaamisaahib@gmail.com (E.U.H.M.); syeda_sam@hotmail.com (S.S.W.); maria00415@yahoo.com (M.I.); maria-bukhari@hotmail.com (M.A.); b.budde@uni-koeln.de (B.B.); wolfgang@fhoe.de (W.H.); janine.altmueller@uni-koeln.de (J.A.); hthiele@uni-koeln.de (H.T.); nuernberg@uni-koeln.de (P.N.); 2Human Molecular Genetics Laboratory, Health Biotechnology Division, National Institute for Biotechnology and Genetic Engineering (NIBGE) College, PIEAS, Faisalabad 38000, Pakistan; saadia.m.saadi@gmail.com (S.M.S.); hychaudhary@gmail.com (H.Y.); ayazgenetics@gmail.com (A.K.); jameel.biotech@gmail.com (M.J.); sherazkhanhu@gmail.com (S.K.); tariqpalai@gmail.com (M.T.); shahid_baig2002@yahoo.com (S.M.B.); 3Neurochemicalbiology and Genetics Laboratory (NGL), Department of Physiology, Faculty of Life Sciences, Government College University, Faisalabad 38000, Pakistan; ghulamhussain@gcuf.edu.pk; 4Institute of Biochemistry I, Medical Faculty, University of Cologne, 50931 Cologne, Germany; stefan.hoening@uk-koeln.de; 5University Institute of Biochemistry and Biotechnology (UIBB), PMAS-Arid Agriculture University, Rawalpindi 46000, Pakistan; uzma.abdullah@uaar.edu.pk; 6Zentrum Medizinische Genetik, Medizinische Universität, 6020 Innsbruck, Austria; sigrid.tinschert@i-med.ac.at; 7Centre for Biotechnology and Microbiology, University of Swat, Swat 19130, Pakistan; zafaralibiotech@gmail.com; 8Department of Biological and Biomedical Sciences, Aga Khan University, Karachi 74000, Pakistan; ambrinfatimach@yahoo.com; 9Bilecik Şeyh Edebali University, Molecular Biology and Genetics, Gülümbe Campus, Bilecik 11230, Turkey; emrahkaygusuz@gmail.com; 10Department of Biotechnology, Kinnaird College for Women, Lahore 54000, Pakistan; iram.anjum@kinnaird.edu.pk; 11Pakistan Science Foundation (PSF), 1- Constitution Avenue, G-5/2, Islamabad 44000, Pakistan; 12Center for Molecular Medicine Cologne (CMMC), University of Cologne, Faculty of Medicine and University Hospital Cologne, 50931 Cologne, Germany

**Keywords:** MCPH, Seckel syndrome, modifier alleles, primordial dwarfism, impaired splicing, supernumerary centrosomes

## Abstract

Congenital microcephaly is the clinical presentation of significantly reduced head circumference at birth. It manifests as both non-syndromic—microcephaly primary hereditary (MCPH)—and syndromic forms and shows considerable inter- and intrafamilial variability. It has been hypothesized that additional genetic variants may be responsible for this variability, but data are sparse. We have conducted deep phenotyping and genotyping of five Pakistani multiplex families with either MCPH (*n* = 3) or Seckel syndrome (*n* = 2). In addition to homozygous causal variants in *ASPM* or *CENPJ*, we discovered additional heterozygous modifier variants in *WDR62, CEP63, RAD50* and *PCNT*—genes already known to be associated with neurological disorders. MCPH patients carrying an additional heterozygous modifier variant showed more severe phenotypic features. Likewise, the phenotype of Seckel syndrome caused by a novel *CENPJ* variant was aggravated to microcephalic osteodysplastic primordial dwarfism type II (MOPDII) in conjunction with an additional *PCNT* variant. We show that the *CENPJ* missense variant impairs splicing and decreases protein expression. We also observed centrosome amplification errors in patient cells, which were twofold higher in MOPDII as compared to Seckel cells. Taken together, these observations advocate for consideration of additional variants in related genes for their role in modifying the expressivity of the phenotype and need to be considered in genetic counseling and risk assessment.

## 1. Introduction

Microcephaly is characterized by a reduced occipitofrontal circumference (OFC) of <−3 standard deviation (SD) based on the age-, gender- and ethnicity-matched mean. It affects 2–3% of the population globally [1] and manifests in cognitive or neurological dysfunction that requires constant medical care. Microcephaly may be congenital (primary) or acquired postnatally (secondary). Primary microcephaly occurs as isolated (without any further features), non-syndromic (with neurological or psychiatric features but without further major morphologic or functional anomalies) and syndromic (in combination with brain malformations and/or other major morphological or functional abnormalities), such as Cohen syndrome (MIM #216550), Cornelia de Lange syndrome (MIM #122470), Nijmegen breakage syndrome (MIM #251260), Smith–Lemli–Opitz syndrome and many more [2].

Primary non-syndromic microcephaly makes a distinct subclass, termed as microcephaly primary hereditary (MCPH (MIM #251200)) or autosomal recessive primary microcephaly. It is known to be associated with 28 different genes, the latest being RRP7A, which encodes a novel component of ribosome biogenesis and was identified in a large Pakistani family [3,4,5]. MCPH has been clinically defined as a disorder of prenatal onset, non-progressive intellectual disability (ID), lack of major brain malformations or major birth defect in a non-central nervous system organ [6]. On the contrary, Seckel syndrome (MIM #210600 for SCKL1) is an allelic disorder of MCPH (shared genetic architecture) that features a short stature and characteristic facies with a prominent nose in addition to congenital microcephaly and ID. Ten genes have been reported for Seckel syndrome so far, and four of them (*CENPJ, CEP152, CDK5RAP2* and *CEP63*) are also involved in MCPH [7]. Microcephalic osteodysplastic primordial dwarfism (MOPD) II manifests features overlapping with Seckel syndrome but these patients feature a characteristic skeletal dysplasia, too. MOPD is known to be caused by homozygous loss-of-function mutations in a particular gene, *PCNT* [8]. For classical MCPH and Seckel syndrome, both intra- and interfamilial clinical variability (even for the same variant) have been frequently observed [9,10], which makes it difficult to infer a genotype–phenotype correlation. Although other genomic loci had been considered to contribute to the variability, they were less tangible through traditional genetic approaches.

With the latest trends and down pricing of next-generation sequencing technologies, it is now possible to identify variants in other genes that are not pathogenic on their own but nevertheless influence the phenotypic outcome of the primary causal variants, called genetic modifiers [11]. Genetic modifiers contribute towards phenotypic variability and penetrance and are thus important in precise diagnostic, prognostic, therapeutic and overall patient management strategies [12]. Such variants or genetic modifiers are being increasingly observed in neurodevelopmental disorders [13,14,15,16] but they have been rarely reported for microcephaly and associated syndromes [17]. Interestingly, they have a pronounced appreciation in mouse models of microcephaly [18,19].

The application of massively parallel sequencing of multiplex families offers a unique opportunity to identify the modifying variants in a similar genetic background. Here, we propose that heterozygous variants of *WDR62, CEP63, RAD50* and *PCNT* contribute to additional neurological and extra-neurological abnormalities in affected siblings of the families manifesting MCPH or Seckel syndrome.

## 2. Materials and Methods

### 2.1. Clinical Manifestations

We performed deep phenotyping and genotyping of five Pakistani multiplex families presenting 28 patients segregating MCPH (three families) or Seckel syndrome (two families) (Table 1, Figure 1A,B). We recruited these families from different areas of Punjab, Pakistan, with prior written consents from the parents of all the families.

In each MCPH family, we noted microcephaly of varying degree (−9 to −15 SD), slopping forehead and articulation difficulties (Figure 1B and Table 1). Intriguingly, some of the kins showed additional phenotypic features, including joint contractures of both elbows and hands, drooling, a short stature (family 1; V-1 and V-9), seizures, hyperactive locomotion, dwarfism (family 2; V-1 and V-2), seizures, aggressive behavior and severe forms of ID (family 3; IV-1 and IV-2) (Table 1 and Figure 1B). Similarly, patients of family 4 (V-6, V-7 and V-8) and 5 (IV-1 and IV-2) featured Seckel syndrome as indicated by a beaky and protruding nose, hypopigmentation (family 4, V-7 only), malocclusion (family 5, IV-1 only) and short stature (Table 1 and Figure 1B), whereas other affected members of family 4 (VI-2, VI-3 VI-4 and VI-5) also manifested mild ID, drooling, clinodactyly of toes and brachydactyly of fingers and toes evincing MOPDII (Table 1 and Figure 1B). Two patients of family 4, one from each loop, were subjected to radiographic analysis, which indicated bilateral clinodactyly of the 4th and 5th toes only in patient VI-4, manifesting the MOPDII phenotype, and it was absent in patient V-7, showing features of Seckel syndrome (Figure 2).

### 2.2. Next-Generation Sequencing

To reveal the causative variants, we selected the affected siblings showing the most severe phenotypic abnormalities for targeted sequencing of a microcephaly gene panel (family 2: V-1, family 3: IV-1, family 4: V-6 and VI-2) or whole-exome sequencing (family 1: V-9 and family 5: IV-1). The complete list of genes included into the panel and the process of targeted sequencing has been published elsewhere [10]. For whole-exome sequencing (WES), we used the Agilent (Santa Clara, CA, USA) version 6 enrichment kit and Illumina HiSeq 4000 sequencing system (paired end reads, 2 × 75bp). The detailed procedure for WES has been described elsewhere [10]. Variant calling and interpretation was performed with the help of our in-house VARBANK database and analysis platform (http://varbank.ccg.uni-koeln.de (accessed on 9 May 2021)).

### 2.3. Linkage Analysis

For family 4, we genotyped the DNA of the four affected members (V-6, V-8, VI-2 and VI-3), indicated by asterisks in Appendix A, using the HumanCoreExome 24 v.1.1 BeadArray (Illumina; San Diego, CA, USA) according to the manufacturer’s instructions. Linkage analysis was performed assuming autosomal recessive inheritance, full penetrance, consanguinity and an allelic frequency of 0.0001. The procedure of data handling and evaluation has been described earlier [20]. The detailed procedure of the statistical analysis is given in Appendix A. To fine-map the *PCNT* locus on chromosome 21, we genotyped six neighboring microsatellite markers with the DNA of ten family members (indicated by asterisks in Appendix A). HaploPainter v.1.043 was used to construct the haplotypes [21]. 

### 2.4. Sanger Sequencing

Co-segregation studies were performed by Sanger sequencing, including all family members for which a DNA sample was available.

### 2.5. In Silico Analyses of Identified Variants

Pathogenicity of the identified variants were predicted by several in silico tools like the ACMG classification system, Mutation Taster, Provean, Polyphen-2, CADD, PANTHER (Paul Thomas, CA, USA), PhD-SNP (Bologna Biocomputing Group, Bologna, Italy), SIFT (Pauline Ng, Genome Institute of Singapore, Singapore), SNAP (Yana Bromberg, University of Columbia, Columbia, NY, USA), Meta SNP (Emidio Capriotti, University of Alabama at Birmingham, Birmingham, AL, USA/Yana Bromberg, Rutgers University, New Brunswick, NJ, USA), MuPro (University of California, Irvine, CA, USA), SNPs&GO (Rita Casadio and Emidio Capriotti, University of Bologna, Italy) and MetaDome (Christian Gilissen, Radboud University Medical Center, Nijmegen, The Netherlands). To investigate the conservation status of the mutated sites, reference sequences were retrieved either from UniProtKB (https://www.uniprot.org/, (accessed on 9 May 2021)) or NCBI (https://www.ncbi.nlm.nih.gov/ (accessed on 9 May 2021)) and were aligned by Clustal Omega (EMBL-EBI, Wellcome Trust Genome Campus, Hinxton, Cambridge, UK). 

### 2.6. Reverse Transcription PCR

To explore the possible consequences of the *CENPJ* missense variant on splicing accuracy, RNA was extracted from the patient’s blood using the PAXgene blood RNA system (QIAGEN, Hilden, Germany) and subsequently converted to cDNA as described previously [22]. In brief, RT-PCR was performed using forward 5′-AGCCACTTGAACCACTGAAC-3′ and reverse 5′-CAGTCTGGTCAGGAAACGTG-3′ primers to amplify the relevant portion of *CENPJ*. Amplicons were resolved on a 2% agarose gel along with a 1 kb plus DNA ladder (10787018, Thermo Fisher Scientific, Waltham, MA, USA) as the size marker. Each band was Sanger sequenced separately to determine the effects on splicing. 

### 2.7. Immunocytochemistry

To investigate the effects of the *CENPJ* missense variant identified in families 4 and 5, primary dermal fibroblasts were cultured from skin biopsies obtained from the affected individuals (V-8 and VI-2) of family 4, as described elsewhere [23]. For immunofluorescence, primary fibroblasts were grown on 12 mm coverslips with a maximum confluency of 70%. The fixation of cells was performed using ice-cold methanol for 10 min at −20 °C. After 10 min, cells were incubated with PBS (1×) followed by the blocking buffer PBG (solution composition reported previously [20]) for 15 min. For immunofluorescence, primary antibodies—mouse anti-CENPJ, 1:25 [24], and rabbit anti PCNT, 1:1000 (Abcam, ab4448)—were applied to the cells and incubated at 4 °C overnight. Alexa Fluor 488 donkey anti-mouse IgG (Invitrogen, A21202) and Alexa Fluor 568 goat anti-rabbit IgG (Invitrogen, A11011) were used as secondary antibodies, each at a dilution of 1:1000, along with 4′,6-diamidin-2′-phenylindol (DAPI) (MilliporeSigma, Saint Louis, MO, USA, D9564) for staining of DNA. The slides were observed under a confocal microscope (Leica Microsystems, Wetzlar Germany, LSM TCS SP5) to capture images. 

### 2.8. Immunoblotting

For immunoblotting, cultured cells from the patients and a control were lysed in lysis buffer (50 mM Tris–HCl, pH 7.5, 150 mM NaCl, 1% Nonidet P-40, 0.5% Na-deoxycholate, 0.1% SDS) supplemented with a proteinase-inhibitor cocktail (MilliporeSigma, Saint Louis, MO, USA, P8340). After denaturing the samples at 95 °C for 8 min in SDS sample buffer, the proteins were finally resolved in 10% SDS-PAGE and blotted onto nitrocellulose membrane (PROTRAN, Sigma Aldrich, Germany). The membrane was incubated overnight at 4 °C with Rabbit anti CPAP, 1:500 (Proteintech Group, Inc, Rosemont, IL, USA, 11517-1-AP) as primary antibody along with rat anti α-tubulin (Y/L1/2) [25], at a dilution ratio of 1:5 for the loading control. Secondary antibodies, anti-rabbit IgG peroxidase conjugate (MilliporeSigma, Saint Louis, MO, USA, A6154) and anti-rat IgG peroxidase conjugate (MilliporeSigma, Saint Louis, MO, USA, A5795) were applied at a dilution ration of 1:10,000, followed by developing the blots using an enhanced chemiluminescence system.

## 3. Results

Family 1 is a very large consanguineous family with eight affected members having microcephaly of −10 to −15 SD accompanied by mild to severe ID (Figure 1A,B and Table 1). Sequencing data analysis of family 1 revealed a nonsense mutation, NM_018136. 4:c. 9601C>T; p.(Gln3201*), of *ASPM* (Table 1). A CADD score of 41 was obtained for this variant and it was placed in the category of pathogenic variants (PVS1, PM2, PP3) by the ACMG classification system. We found it neither in gnomAD nor in dbSNP build 153 (Table 2). This variant was segregating, as expected, a recessive inheritance pattern (Figure 1A and Appendix A). Additional phenotypic presentation of the contracted joints, drooling and short stature, which were observed only in a few affected members of this family, prompted us to look for additional potentially disease-causing variants. As a result, we found a heterozygous missense variant, NM_001083961.1:c.3316A>G;p.(Ser1106Gly), of *WDR62* (Table 1). Interestingly, this variant is not catalogued in Iranome and the Greater Middle Eastern Variome. It is, however, present in gnomAD (with two heterozygous alleles) and dbSNP build 153 (rs1389367700) (Table 2). Based on this ultralow allele frequency, we investigated this variant further. Sanger sequencing revealed that this variant was found only in patients showing additional features of joint contractures, short stature and drooling (Figure 1A and Appendix A, Table 1). The variant p.(Ser1106Gly) does not reside in any of the 15 WD repeats of WDR62 yet shows its deleterious nature by PhD-SNP, SIFT, SNAP, Meta SNP and MuPro (Table 2). According to the ACMG classification system, it is considered as a variant of uncertain significance (VUS). This position is not conserved in mammals in general (Appendix A); however, it is strictly conserved among primates (Appendix A). Notably, this variant could be detrimental and impair the function of WDR62 because Ser1106 is reported to be phosphorylated during mitosis [26].

Family 2 is a five-generation consanguineous family with five affected members, two of them deceased for unknown reason, manifesting microcephaly of −11 to −15 SD, moderate ID and aggressive behavior (Figure 1A,B and Table 1). Sequence data analysis of this family revealed a frameshift variant, NM_018136. 4:c.719_720delCT;p.(Ser240Cysfs*16), in *ASPM*. As presumed, all the tested patients were homozygous for this variant (Figure 1A and Appendix A, Table 1). Because clinical manifestation of the proband subjected to gene panel sequencing was severe (Table 1), the possibility of additional variants was explored. We found a missense variant, NM_025180. 3:c.1261A>T;p.(Thr421Ser), of *CEP63* that was subsequently confirmed in his younger sister (V-2), manifesting a similarly severe clinical presentation (Figure 1A and Table 1). Their asymptomatic brother (V-3) was a heterozygous carrier of both the aforementioned variants in *ASPM* and *CEP63*. The *CEP63* variant was inherited from their asymptomatic father (IV-6), whereas the mother was homozygous for the wild-type allele (Figure 1A and Appendix A). Interestingly, investigation of a second loop of the same family showed absence of the *CEP63* variant in affected (IV-1) as well as in phenotypically normal members (IV-3 and IV-4), even though their mother was heterozygous for this variant (Figure 1A). The variant was predicted to be disease-causing, deleterious and probably damaging, and decreasing the protein stability by the *in silico* tools Mutation Taster (score = 58), Provean (score = −2.830), Polyphen-2 (score = 1.000) and MuPro (DDG = −0.54948253), respectively; however, it was classified as VUS (PM2, BP1) by the ACMG classification system (Table 2). The pathogenic potential of this variant was further strengthened by its absence in databases of genomic variants like gnomAD, GME variome, Iranome and dbSNP build 153 (Table 1). Furthermore, multiple alignments of Cep63 peptides, spanning the mutant residue, revealed that threonine at position 421 of the human protein is highly conserved throughout the vertebrate lineage (Appendix A).

Family 3 is consanguineous, having three affected female siblings showing HC of -9 to −13 SD with phenotypic variability in the manifestation of seizures, ID and behavior (Figure 1A,B and Table 1). Variant filtration identified a nonsense mutation, NM_018136.4:c.9492T>G;p.(Tyr3164*), of *ASPM*, homozygous in all affected members (Figure 1A and Appendix A, Table 1). Two affected members (IV-1 and IV-2) of this family presented severe ID accompanied by aggressive behavior compared to their younger sister (IV-3), who showed mild ID and normal behavior (Table 1). Considering the heterogeneity of the clinical features, we extended the search for additional disease-causing variant(s) in other genes. To this end, we identified a heterozygous missense variant, NM_005732. 3:c.3643C>G;p.(Leu1215Val), of *RAD50* inherited from their asymptomatic mother (III-2) (Figure 1A and Appendix A). This variant is not listed in any of the aforementioned public databases of genomic variations and is predicted to be pathogenic by Mutation Taster, Polphen-2, PANTHER, PhD-SNP, SIFT, SNAP, Meta SNP, MuPro, MetaDome and CADD (score = 23.5), albeit likely benign (PM1, PM2, BP1, BP4) by the ACMG (Table 2). We considered it a potential modifier because this variant was only carried by affected members with severe phenotypic manifestations, IV-1 and IV-2, whereas the affected member IV-3 with a milder phenotype did not carry this variant (Figure 1A,B). Furthermore, the mutated residue is highly conserved within a large group of organisms, including vertebrates, insects, plants and even yeast—only worms contain leucine at this position (Appendix A).

Family 4 is a multigenerational family with nine affected members born to two consanguineous couples and exhibiting major phenotypic differences among patients of both loops (Figure 1A,B). Affected siblings in loop 1 (right side) showed typical features of Seckel syndrome whereas those of loop 2 (left side) showed more severe clinical features reminiscent of MOPDII (Figure 1A,B and Table 1). Affected members of loop 1 showed moderate ID, HC of −7 to −8.5 SD, short stature and protruding noses. In addition, two of the three affected individuals of loop 1 showed hypopigmentation (Table 1 and Figure 1B, white arrowheads). Affected members of loop 2 showed microcephaly of HC −10 to −13 SD, short stature, mild ID, excessive drooling, speech impairment, clinodactyly of only toes and brachydactyly of both fingers and toes (Figure 1B and Figure 2, Table 1), suggesting a different cause or that a genetic modifier might confer additional features. Gene panel sequencing of two patients, one of each loop, demonstrated a novel homozygous *CENPJ* variant (NM_018451.4:c.3586G>A;p.(Asp1196Asn)) in both patients that were later confirmed in all affected siblings of both loops by Sanger sequencing (Figure 1A and Appendix A). The involvement of the missense variant of *CENPJ* in the disease etiology was further corroborated by linkage analysis that highlighted a region on chromosome 13 with a maximum LOD score of 4.3 (Appendix A). The homozygous segment of 2.63 Mb was flanked by the markers rs7981441 (24,075,006bp; GRCh38.p12) and rs9551309 (26,712,326bp; GRCh38.p12), thus harboring *CENPJ* (Appendix A). 

Screening for additional variants revealed a previously reported heterozygous nonsense variant of *PCNT* (NM_006031.5;c.5767C>T;p.Arg1923*) in the patients of loop 2, suggesting a modifier role for *PCNT* (Figure 1A and Appendix A, and Table 2). Sanger sequencing demonstrated that the *PCNT* variant was inherited from the mother (V-2) and carried by all the affected members of this loop, whereas this variant was absent from all individuals of loop 1 (Figure 1A). Additionally, genotyping of microsatellite markers in selected family members revealed that the *PCNT* variant haplotype (colored in red, Appendix A) was introduced by the unrelated male individual IV-2 and absent from all tested family members of loop 1. The *PCNT* variant obtained a very high CADD score of 42 and was predicted to be pathogenic (PVS1, PM2, PP3, PP5) by ACMG criteria and disease-causing by Mutation Taster (Table 2). 

We considered the missense mutation, NM_018451.4:c.3586G>A;p.(Asp1196Asn), of *CENPJ* causative of Seckel syndrome in family 4 because it was not recorded in any of the genomic variation databases and predicted to be pathogenic with a CADD score of 28.1, likely pathogenic (PM1, PM2, PP3, PP5) by the ACMG classification system and disease-causing by Mutation Taster, PROVEAN, Polphen-2, PhD-SNP, SIFT, SNAP, Meta SNP, MuPro and MetaDome (Table 2). Furthermore, the amino acid position 1196 of CENP-J is strictly conserved from fungi to vertebrates, advocating for an indispensable role of this aspartic acid residue (Appendix A). RT-PCR analysis of a fragment spanning the variant and neighboring exons using patient and control mRNA revealed that the presumable missense mutation c.3586G>A (p.(Asp1196Asn)) of *CENPJ* at least partially also results in aberrant splicing by activating a cryptic splice donor site located within exon 14. We noted two bands of reduced intensity as compared to the single band of expected size (372 bp) seen in the control (Figure 3A). One band of the mutant sample was of the same size as that of the control while the second band was considerably smaller (~294 bp) (Figure 3A). Sanger sequencing of the wild-type PCR product resulted in the expected transcript sequence ranging from exon 13 to exon 16 (Figure 3B,C, upper panels). Sequencing of the upper mutant band revealed an identical sequence with the only exception of the mutant nucleotide c.3586G>A (Figure 2, lower left panel). Sequencing of the lower band, however, revealed an in-frame deletion of 78 bp (c.3541_3618del) covering the 3′ region of exon 14 (Figure 3B, lower right panel and Figure 3C, lower panel). Evidently, the mutation c.3586G>A of *CENPJ* results in two different transcripts that are nearly equally represented, a full-length one encoding a peptide with only one altered amino acid, p.(Asp1196Asn), and a shorter one that encodes a peptide which lacks 26 amino acids, p.(Val1181_Val1206del).

These data intrigued us to follow the consequences of mutant CENP-J at the cellular level. Immunofluorescence analysis in primary fibroblasts of wild-type origin revealed bright staining of CENP-J at the centrosome, which is marked by pericentrin (Figure 3D, white arrow). Contrarily, primary fibroblasts derived from two different patients of family 4—one carrying only the *CENPJ* mutation (V-8) and another carrying the *CENPJ* and the *PCNT* mutation (VI-2)—showed faint staining of CENP-J, marked by white arrow heads (Figure 3D). These findings were corroborated by immunoblotting where protein lysates of the patient-derived primary fibroblasts showed faint immunoreactivity of CENP-J as compared to the wild type (Figure 3E, upper panel, and Appendix A). Equal concentrations of lysates were loaded as evident by the immunoreactivity of α-tubulin (Figure 3E, lower panel, and Appendix A). In addition to the compromised stability of CENP-J, we also observed a significant number of patient interphase cells with supernumerary centrosomes (Figure 3D). On average, 7.33% of cells derived from the patient carrying only the *CENPJ* mutation showed supernumerary centrosomes, whereas more than twice as many (15.66%) of the cells derived from the patient carrying both mutations showed them (Figure 3F). Notably, only 1.66% of the wild-type cells showed extra centrosomes (Figure 3F). Based on this data, we speculate that the *PCNT* variant exacerbates cellular anomalies caused by CENP-J dysfunction, consequently worsening the clinical phenotype in VI-2.

Family 5 is a consanguineous family with two affected male siblings showing typical features of Seckel syndrome, such as a prominent nose, HC-11 to -12 SD and short stature (Figure 1A,B and Table 1). Whole-exome sequencing, performed in one affected member, revealed the same *CENPJ* missense variant (NM_018451.4:c.3586G>A;p.(Asp1196Asn)) that had been identified in family 4 (Figure 1A,B and Appendix A, Table 1 and Table 2). The identification of the same *CENPJ* variant in a second unrelated family with Seckel syndrome provides strong evidence for its causal role in the disease etiology. Interestingly, exome data analysis did not show any variation of *PCNT* in family 5.

## 4. Discussion

Here, we have provided evidence that heterozygous missense variants of *WDR62, CEP63* and *RAD50* aggravate the phenotype of MCPH and a *PCNT* nonsense variant exacerbates the severity of Seckel syndrome features. Multiple lines of evidence advocate for their potential to act as genetic modifiers; this includes absence of any record or report of the rare alleles in the literature or databases of genomic variations, high pathogenicity scores as calculated by several in silico tools, high conservation of the affected amino acid positions, and last but not least, the known involvement of these genes in the neurodevelopmental disorders, which are discussed below in a continuum of the respective modifiers. 

The genes identified with modifier variants in this study have previously been found dysfunctional in non-syndromic primary microcephaly and its syndromic forms. Mutations in *WDR62* were reported to cause primary microcephaly with or without severe brain malformations (lissencephaly and pachygyria) [27,28]. A nonsense mutation in *CEP63* was reported to cause Seckel syndrome in a Pakistani family, with three affected members showing microcephaly and reduced height [29]. For this reason, short stature in our patients could be explained by the combined effect of this heterozygous modifying variation of *CEP63* together with the homozygous 2 bp deletion in *ASPM*. Nonsense and frameshift variants of *RAD50* were previously reported in families afflicted with Nijmegen breakage syndrome-like disorder (MIM #613078). Patients with this disorder feature microcephaly, ID, a characteristic ‘bird-like’ face and short stature [30]. Interestingly, mutations in *PCNT* were found to be associated with MOPDII [8,31]. Notably, many of the encoded proteins interact with each other and form a crucial complex necessary for centriole biogenesis; e.g., Cep63, Asp homolog, also known as ASPM, and WDR62 recruit CENP-J at the site of centriole biogenesis [7]. Therefore, it could be speculated that modifying variants may affect the structure and/or function of the complex, thereby resulting in a more severe phenotype. A critical role of the WDR62 mutation p.(Ser1106Gly), identified in family 1, is highly convincing as Ser1106 has been reported to be a key site of phosphorylation in mitosis [26]. Additionally, the conservation constraint on Ser1106 in primates could also implicate that serine at this position might be subject to positive selection of this gene attributing some primate-specific function. 

hRAD50 is a well-known component of the hRAD50 MRN (Mre11, Rad50 and Nbs1) complex with a well-established role in double-strand break (DSB) repair [32]. Our patients, carrying a variation of *RAD50*, showed rare and irregular episodes of seizures, aggressive behavior and severe ID. None of these features were observed in the affected sibling (IV-3) who did not carry the *RAD50* variation (Table 1). Furthermore, the variant p.(Leu1215Val) resides within an Ala/Asp-rich domain (DA-box) highly conserved across vertebrates. 

CENP-J mutation p.(Asp1196Asn) is located in its T complex protein 10 (TCP) domain (residues 1159–1337), which interacts with STIL [33,34] and plays a crucial role during centrosome biogenesis by mediating the tethering of pericentriolar material [34]. Another mutation of *CENPJ*, NM_018451.4:c.3704A>T;p.(Glu1235Val), reported in a Pakistani MCPH patient [35], lies close to our mutation and has been shown to impair this interaction, thus inhibiting centriole biogenesis [33]. Therefore, we speculate that the mutation identified by us may also impair the interaction with STIL and compromise the tethering ability of CENP-J in general. Notably mutant CENP-J encoded by the second mutant transcript lacks 26 amino acids (p.(Val1181_Val1206del)), spanning a region of the TCP 10 domain and could also show detrimental effects on the interaction with STIL. Furthermore, aspartic acid at position 1196 is a highly conserved residue and, due to its negative charge, most likely participates in protein/protein interaction.

Interestingly, homozygous pericentrin mutation p.Arg1923* was originally described as a cause of MOPDII [8]. If the mutant protein would be formed by escaping from nonsense mediated mRNA decay, it would result in a loss of two helical strands, loss of nek2 interaction (residues 2983–3246) and loss of calmodulin-binding (residues 3195–3208), which should result in severe consequences for the protein function in centrosome arrangement. Therefore, we have seen severe centrosomal defects in cells carrying mutations in *CENPJ* and *PCNT*. On a background of biallelic *CENPJ* mutations, haploinsufficiency of *PCNT* may become overt and result in a more severe clinical picture, as seen in some of our patients of family 4 (Figure 1B and Table 1).

To the best of our knowledge, there has been only one publication reporting the contribution of a genetic modifier to the phenotype of MCPH. In addition to a homozygous stop mutation in *WDR62*, NM_001083961.1:c.1605dupT;p.(Glu536*), originally reported as c.1605_1606insT, the MCPH patient also carried a duplication of the chromosomal segment 17q25-qter and a missense mutation, c.3361T>G;p.Phe1121Val, of *TBCD* on the non-duplicated allele. More severe phenotypic manifestations were observed in this patient as compared to one carrying only the homozygous *WDR62* stop mutation [17], which was attributed to the modifying effects of *TBCD*. In case of Seckel syndrome, only a digenic inheritance pattern has been reported, with heterozygous mutations of *CDK5RAP2* and *CEP152* [36]. Another report showed apparently digenic triallelic inheritance in patients manifesting MCPH, with the following combinations of genes: *WDR62/CDK5RAP2, ASPM/WDR62,* and *WDR62/CEP135*. Unfortunately, the authors did not present any data for the presumptive modifying effects of the additional variants on the phenotype. Furthermore, they did not find any modifying effects of *wdr62* or *aspm* ablations in zebrafish upon knocking out *casc5/knl1*; merely, the quadriallelic ablation of *wdr62* and *aspm* showed primary microcephaly in zebrafish [37]. Hence, these data lay a foundation for future studies to investigate the modifying variants in MCPH cases. Importantly, the phenotypic severity, corroborated by the sub-cellular anomalies confirmed by our experiments, models an efficient strategy for future studies for validation of the modifying variants at the functional level.

## 5. Conclusions

The data presented in our study are unique for the fact that our multiplex pedigrees show intrafamilial phenotypic and genotypic differences, thus providing a solid basis to investigate the role of potentially modifying variants. We have provided compelling evidence of the pathogenic nature of these variants, yet further functional analyses will provide deeper insights into the pathomechanisms involving more than one mutated gene. Based on our data, we recommend clinicians and researchers to subject the DNA of the most severely affected members of larger families to comprehensive genomic analyses, which may help to identify supplementary disease-modifying gene variants in addition to the disease-causing ones.

## Figures and Tables

**Figure 1 genes-12-00731-f001:**
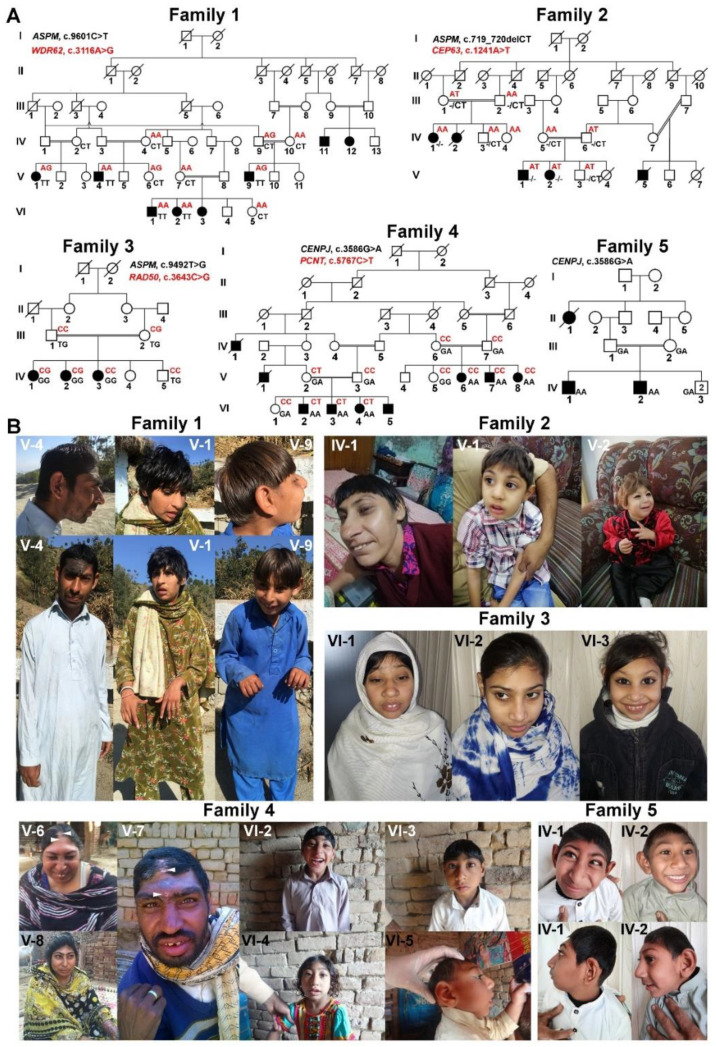
Pedigrees and clinical features of patients. (**A**) Pedigrees of families manifesting primary microcephaly and Seckel syndrome carrying homozygous loss-of-function mutations in *ASPM* and *CENPJ* (shown in black beneath symbols), respectively, and heterozygous mutations in the proposed modifier genes (shown in red above the symbols). Affected members of family 5 were found mutated only for *CENPJ* but not for *PCNT*. (**B**) Photos of the selected individuals belong to the five families. Notably, joint contractures are clearly visible in photos of the affected members, V-1 and V-9, of family 1. Family 4 (V-6, V-7 and V-8) and family 5 feature Seckel syndrome whereas patients in the left loop of family 4 (VI-2, VI-3, VI-3 and VI-5) are clinically diagnosed with MOPDII. White arrowheads shown on the forehead areas of V-6 and V-7 of family 4 indicate hypopigmentation.

**Figure 2 genes-12-00731-f002:**
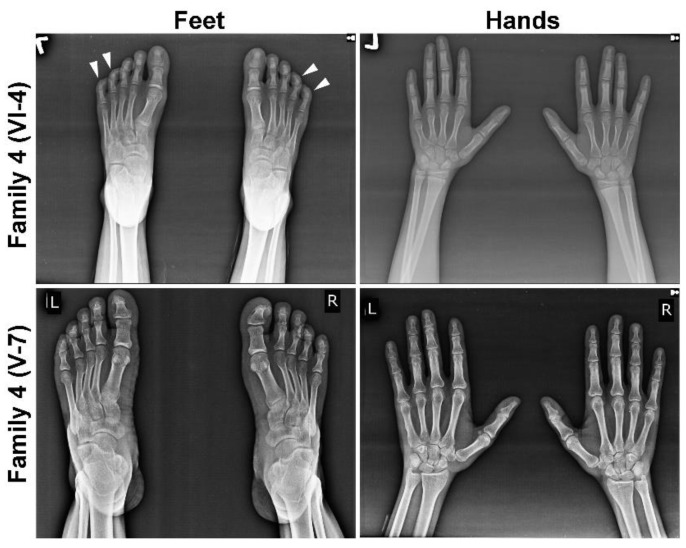
Radiographic images of selected members of family 4. Radiographs of hands and feet of MOPDII patient VI-4 (upper panel) and Seckel patient V-7 (lower panel). Bilateral clinodactyly of the 4th and 5th toes is seen only in MOPDII patients, indicated by white arrow heads.

**Figure 3 genes-12-00731-f003:**
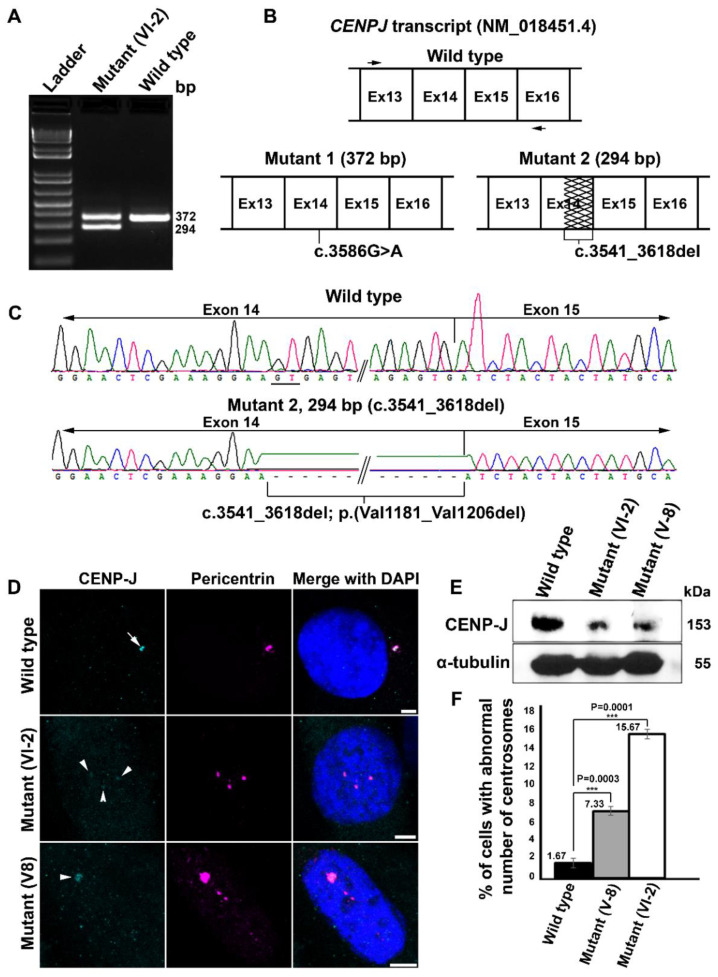
Investigation of *CENPJ* transcript and protein in family 4. (**A**) Electropherogram (2% agarose) of amplicons covering portion of *CENPJ* obtained from cDNA of patient VI-2 of family 4 along with the wild type. (**B**) Schematic of partial region of *CENPJ* wild-type transcript (top) showing the locations of the primers (black arrows) used to amplify the cDNA. Lower left panel shows the schematic of *CENPJ* transcript with a size comparable to that of the wild type and carrying the missense mutation. Lower right panel indicates the schematic of the mis-spliced transcript, with a 78-bp deletion (shown by zigzag lines) in exon 14. (**C**) Sanger traces of the amplicons obtained from mutant and wild-type cDNA. The top panel shows traces of the wild type and the lower panel belongs to the smaller, mis-spliced mutant, showing deletion of 78 bp. The position of the cryptic splice donor site (GT) is underlined in wild-type traces. (**D**) Confocal microscopy images of the primary fibroblasts derived from different patients (V-8 and VI-2) of family 4. Localization of CENP-J is indicated by white arrows (wild type) and arrowheads (mutants). Reduced immunoreactivity is evident in both mutants as compared to the wild type. Scale bar 5 µm. (**E**) Immunoblotting showing the relative quantity of CENP-J (upper panel) in wild-type and patient-derived primary fibroblasts. The internal control α-tubulin is shown in the lower panel. (**F**) Percentage cells with supernumerary centrosomes in wild-type and patient-derived primary fibroblasts of family 4. Exact data points are shown on the top left side of the respective bar. Error bars depict the standard deviation (SD). The *p* values are 0.0003 for mutant V-8 and 0.0001 for mutant VI-2 (Student’s *t* test); three experiments were considered where 100 cells were accounted for each experiment.

**Table 1 genes-12-00731-t001:** Clinical and molecular presentations of MCPH and Seckel syndrome families carrying mutations in the main genes that induce the phenotypes and the modifier genes that result in the enhancement of the phenotypes.

**(a)**												
	**Family 1**						**Family 2**			**Family 3**		
**Patient ID**	**VI-1**	**VI-2**	**VI-3**	**V-4**	**V-1**	**V-9**	**IV-1**	**V-1**	**V-2**	**IV-1**	**IV-2**	**IV-3**
**Gene 1**												
Gene name	*ASPM*	*ASPM*	*ASPM*	*ASPM*	*ASPM*	*ASPM*	*ASPM*	*ASPM*	*ASPM*	*ASPM*	*ASPM*	*ASPM*
Zygosity	Homo	Homo	Homo	Homo	Homo	Homo	Homo	Homo	Homo	Homo	Homo	Homo
cDNA mutation	c.9601C>T	c.9601C>T	c.9601C>T	c.9601C>T	c.9601C>T	c.9601C>T	c.719_720delCT	c.719_720delCT	c.719_720delCT	c.9492T>G	c.9492T>G	c.9492T>G
Protein mutation	p.(Gln3201*)	p.(Gln3201*)	p.(Gln3201*)	p.(Gln3201*)	p.(Gln3201*)	p.(Gln3201*)	p.(Ser240Cysfs*16)	p.(Ser240Cysfs*16)	p.(Ser240Cysfs*16)	p.(Tyr3164*)	p.(Tyr3164*)	p.(Tyr3164*)
**Gene 2**												
Gene name	-	-	-	-	*WDR62*	*WDR62*	-	*CEP63*	*CEP63*	*RAD50*	*RAD50*	-
Zygosity	-	-	-	-	Hetero	Hetero	-	Hetero	Hetero	Hetero	Hetero	-
cDNA mutation	-	-	-	-	c.3116A>G	c.3116A>G	-	c.1241A>T	c.1241A>T	c.3643C>G	c.3643C>G	-
Protein mutation	-	-	-	-	p.(Ser1106Gly)	p.(Ser1106Gly)	-	p.(Thr421Ser)	p.(Thr421Ser)	p.(Leu1215Val)	p.(Leu1215Val)	-
**Measurement**												
Age (years)	12	9	6	23	32	10	34	6	3	18	14	10
Gender	Male	Female	Female	Male	Female	Male	Female	Male	Female	Female	Female	Female
HC (cm)	41	35	34	42	41.5	37	35	37.5	32	41	43	38
HC (SD)	−10	−14.55	−15	−10	−10	−12	−14.55	−11	−15	−10.5	−9	−13
Height (cm)	133	115	105	162	**124**	**99**	145	**93**	**76**	137	145	112
Height (SD)	−2	−3	−2.5	−2	**−7**	**−6**	−3	**−5**	**−5**	−5	−3	−4
**Neurological features**												
ID	Moderate	Mild	Severe	Moderate	Severe	Mild	Moderate	Moderate	Moderate	**Severe**	**Severe**	Mild
Behavior	Aggressive	Aggressive	Aggressive	Aggressive	Aggressive	Aggressive	Aggressive	Aggressive	Aggressive	Aggressive	Aggressive	Normal
Speech impairment	Severe	Severe	Severe	Severe	Severe	Severe	Severe	Moderate	Severe	Mild	Mild	Mild
**Musculoskeletal abnormalities**												
Contractures	-	-	-	-	**Joints (elbow and hands)**	**Joints (elbow and hands)**	-	-	-	-	-	-
Clinodactyly of toes	-	-	-	-	-	-	-	-	-	-	-	-
Clinodactyly of fingers	-	-	-	-	-	-	-	-	-	-	-	-
Brachydactyly of fingers	-	-	-	-	-	-	-	-	-	-	-	-
Brachydactyly of toes	-	-	-	-	-	-	-	-	-	-	-	-
**Others**												
Drooling	-	-	-	-	**Yes**	**Yes**	-	-	-	-	-	-
Locomotion	Normal	Normal	Normal	Normal	Normal	Normal	Normal	**Hyperactive**	**Hyperactive**	Normal	Normal	Normal
Seizures	-	-	-	-	-	-	-	**Yes**	**Yes**	**Yes**	**Yes**	-
Hypopigmentation	-	-	-	-	-	-	-	-	-	-	-	-
Teeth	Normal	Normal	Normal	Normal	Normal	Normal	Normal	Normal	Normal	Normal	Normal	Normal
**(b)**												
	**Family 4**										**Family 5**	
**Patient ID**	**V-6**		**V-7**		**V-8**	**VI-2**		**VI-3**	**VI-4**		**IV-1**	**IV-2**
**Gene 1**												
Gene name	*CENPJ*		*CENPJ*		*CENPJ*	*CENPJ*		*CENPJ*	*CENPJ*		*CENPJ*	*CENPJ*
Zygosity	Homo		Homo		Homo	Homo		Homo	Homo		Homo	Homo
cDNA mutation	c.3586G>A		c.3586G>A		c.3586G>A	c.3586G>A		c.3586G>A	c.3586G>A		c.3586G>A	c.3586G>A
Protein mutation	p.(Asp1196Asn) and p.(Val1181_Val1206del)		p.(Asp1196Asn) and p.(Val1181_Val1206del)		p.(Asp1196Asn) and p.(Val1181_Val1206del)	p.(Asp1196Asn) and p.(Val1181_Val1206del)		p.(Asp1196Asn) and p.(Val1181_Val1206del)	p.(Asp1196Asn) and p.(Val1181_Val1206del)		p.(Asp1196Asn) and p.(Val1181_Val1206del)	p.(Asp1196Asn) and p.(Val1181_Val1206del)
**Gene 2**												
Gene name	-		-		-	*PCNT*		*PCNT*	*PCNT*		-	-
Zygosity	-		-		-	Hetero		Hetero	Hetero		-	-
cDNA mutation	-		-		-	c.5767C>T		c.5767C>T	c.5767C>T		-	-
Protein mutation	-		-		-	p.(Arg1923*)		p.(Arg1923*)	p.(Arg1923*)		-	-
**Measurement**												
Age (years)	32		22		20	8		6	4		10	4
Gender	Female		Male		Female	Male		Male	Female		Male	Male
HC (cm)	44		47		45	39		35	35		35	35
HC (SD)	−8.5		−7		−7.5	−10		−13	−11.5		−12	−11
Height (cm)	139		156		145	107		91	86		114	81
Height (SD)	−4.5		−4		−4	−4		−5	−4		−4	−5
**Neurological features**												
ID	Moderate		Moderate		Moderate	**Mild**		**Mild**	**Mild**		Moderate	Moderate
Behavior	-		-		-	-		-	-		Normal	Aggressive
Speech impairment	Severe		Moderate		Mild	Moderate		Severe	Moderate		Moderate	Moderate
**Musculoskeletal abnormalities**												
Contractures	-		-		-	-		-	-		-	-
Clinodactyly of toes	-		-		-	Yes (bilateral)		Yes (bilateral)	Yes (bilateral)		-	-
Clinodactyly of fingers	-		-		-	-		-	-		-	-
Brachydactyly of fingers	-		-		-	Yes (bilateral)		Yes (bilateral)	Yes (bilateral)		-	-
Brachydactyly of toes	-		-		-	Yes (bilateral)		Yes (bilateral)	Yes (bilateral)		-	-
**Others**												
Drooling	-		-		-	**Yes**		**Yes**	**Yes**		Very rare	-
Locomotion	Normal		Normal		Normal	Normal		Normal	Normal		Normal	Normal
Seizures	-		-		-	-		-	-		Very rare	Normal
Hypopigmentation	**Yes (forehead)**		**Yes (forehead)**		-	-		-	-		-	-
Teeth	Normal		Normal		Normal	Normal		Normal	Normal		Malocclusion	Normal

Note: “-” means absence of the feature; NR, No Record; HC, Head Circumference; ID, Intellectual Disability; cm, Centimeter; SD, Standard Deviation. Second mutation of *CENPJ* shown in family 4 and 5 is based on RT-PCR data, which revealed a second transcript lacking 78 bp (c.3541_3618del) of *CENPJ* and resulting in an in-frame deletion of 26 amino acids, p.(Val1181_Val1206del).

**Table 2 genes-12-00731-t002:** Pathogenicity chart of the mutations identified in Pakistani families.

**(a)**										
**ID**	**Gene**	**Transcript ID**	**Exon**	**cDNA Change**	**Protein Change**	**ACMG Interpretation**	**gnomAD** **Frequency**	**CADD-** **Score**	**Mutation** **Taster**	**Polyphen-2**
**Fam. 1**	*ASPM*	NM_018136.4	23	c.9601C>T	p.(Gln3201*)	Pathogenic (PVS1, PM2, PP3)	-	41	Disease causing (6.0)	NA
	*WDR62*	NM_001083961.1	27	c.3316A>G	p.(Ser1106Gly)	Uncertain significance (PM2)	0.0000325	19.21	Polymorphism (56)	Benign (0.09)
**Fam. 2**	*ASPM*	NM_018136.4	3	c.719_720delCT	p.(Ser240Cysfs*16)	NA	0.0000325	-	Disease causing	NA
	*CEP63*	NM_025180.3	12	c.1261A>T	p.(Thr421Ser)	Uncertain significance (PM2, BP1)	-	25.1	Disease causing (58)	Probably damaging (1.000)
**Fam. 3**	*ASPM*	NM_018136.4	23	c.9492T>G	p.(Tyr3164*)	Pathogenic (PVS1, PM2, PP5)	-	35	Disease causing (6.0)	NA
	*RAD50*	NM_005732.3	24	c.3643C>G	p.(Leu1215Val)	Likely benign (PM1, PM2, BP1, BP4)	-	23.5	Disease causing (32)	Probably damaging (1.000)
**Fam. 4**	*CENPJ*	NM_018451.4	14	c.3586G>A (c.3541_3618del) ^†^	p.(Asp1196Asn) p.(Val1181_Val1206del) ^††^	Likely pathogenic (PM1, PM2, PP3, PP5)	-	28.1	Disease Causing (23)	Probably Damaging (1.000)
	*PCNT*	NM_006031.5	28	c.5767C>T	p.(Arg1923*)	Pathogenic (PVS1, PM2, PP3, PP5)	0.0000122	42	Disease Causing (6.0)	NA
**Fam. 5**	*CENPJ*	NM_018451.4	14	c.3586G>A (c.3541_3618del) ^†^	p.(Asp1196Asn) p.(Val1181_Val1206del) ^††^	Likely pathogenic (PM1, PM2, PP3, PP5)	-	28.1	Disease Causing (23)	Probably Damaging (1.000)
**(b)**										
**ID**	**Gene**	**Provean**	**PANTHER**	**PhD-SNP**	**SIFT**	**SNAP**	**Meta SNP**	**MuPro**	**SNPs&GO**	**MetaDome**
**Fam. 1**	*ASPM*	NA	NA	NA	NA	NA	NA	NA	NA	NA
	*WDR62*	Neutral (−1.216)	NA	Disease causing (0.589)	Disease causing (0.010)	Disease causing (0.505)	Disease causing (0.514)	DDG = −1.6848846 (DECREASE stability)	Neutral	Slightly tolerant (0.93)
**Fam. 2**	*ASPM*	NA	NA	NA	NA	NA	NA	NA	NA	NA
	*CEP63*	Deleterious (−2.830)	NA	Neutral (0.380)	NA	NA	Neutral (0.203)	DDG = −0.54948253 (DECREASE stability)	Neutral	Tolerant (1.33)
**Fam. 3**	*ASPM*	NA	NA	NA	NA	NA	NA	NA	NA	NA
	*RAD50*	Neutral (−2.238)	Disease causing (0.603)	Disease causing (0.591)	Disease causing (0.010)	Disease causing (0.645)	Disease causing (0.654)	DDG = −0.79919509 (DECREASE stability)	Neutral	Intolerant (0.42)
**Fam. 4**	*CENPJ*	Deleterious (−4.841)	NA	Disease causing (0.833)	Disease causing (0.000)	Disease causing (0.685)	Disease causing (0.706)	DDG = −0.82129284 (DECREASE stability)	Neutral	Intolerant (0.33)
	*PCNT*	NA	NA	NA	NA	NA	NA	NA	NA	NA
**Fam. 5**	*CENPJ*	Deleterious (−4.841)	NA	Disease causing (0.833)	Disease causing (0.000)	Disease causing (0.685)	Disease causing (0.706)	DDG = −0.82129284 (DECREASE stability)	Neutral	Intolerant (0.33)

Note: (a) ^†^ This particular cDNA mutation (78 bp deletion) resulted due to aberrant splicing examined by RT-PCT. ^††^ Protein mutation due to a 78 bp in-frame deletion. Allele frequency shown in the column of gnomAD was based on two heterozygous alleles in *WDR62*, one allele of *ASPM* and two heterozygous alleles of *PCNT*. Numbers shown in brackets below the prediction statuses indicate the pathogenicity scores suggested by each tool. Mutations of *ASPM* and *CENPJ* are in homozygous forms whereas *WDR62, CEP63, RAD50* and *PCNT* are in heterozygous states. Value reported under each prediction: PANTHER, PhD-SNP, SNAP and Meta-SNAP (scale is 0 to 1 and more than 0.5 score signifies disease causing); SIFT (positive values: more than 0.5 score shows neutral effects of mutation); and MuPro (a score less than 0 means the mutation decreases the protein stability. NA means data not available. (b) ^†^ This particular cDNA mutation (78 bp deletion) resulted due to aberrant splicing examined by RT-PCT. ^††^ Protein mutation due to a 78 bp in-frame deletion. Mutations of *ASPM* and *CENPJ* are in homozygous forms whereas *WDR62, CEP63, RAD50* and *PCNT* are in heterozygous states. Allele frequencies shown in the column of gnomAD were based on two heterozygous alleles in *WDR62*, one allele of *ASPM* and two heterozygous alleles of *PCNT*. Numbers shown in brackets below the prediction statuses indicate the pathogenicity scores suggested by each tool. Interpretation of values for PANTHER, PhD-SNP, SNAP and Meta-SNAP, scale is 0 to 1 and more than 0.5 score signifies disease causing; SIFT, positive values: more than 0.5 score shows neutral effects of mutation; and MuPro, a score less than 0 means the mutation decreases the protein stability. NA means data not available.

## Data Availability

Data concerning this study is available in the article and Appendix A.

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
