# Peer review of "Modifier Genes in Microcephaly: A Report on WDR62, CEP63, RAD50 and PCNT Variants Exacerbating Disease Caused by Biallelic Mutations of ASPM and CENPJ"

_genes, 2021, doi:10.3390/genes12050731_

Round 1

Reviewer 1 Report

Makhdoom et al., present evidence to consider WDR62, CEP63, RAD50 and PCNT variants to act as modifier loci in microcephaly cases primarily due to variants in ASPM and CENPJ. The data support the conclusions and my comments are minor.

  1. please alter the presentation of the data in Fig. 3F to include the actual data points to help us appreciate the raw data. please explicitly state p values.
  2. what is the predicted protein from the deletion shown in Fig.3C?
  3. I think this is a great opportunity to robustly discuss the need and opportunity to follow up these genetic associations with biological studies to support the bi-alleleic effects. Some of that such work is done here but a call to action for this would be well placed here.

Author Response

Below, we are copying Reviewers comments/questions individually, followed by our responses (in italics).

Reviewer 1:

Comments to the Author

Makhdoom et al., present evidence to consider WDR62, CEP63, RAD50 and PCNT variants to act as modifier loci in microcephaly cases primarily due to variants in ASPM and CENPJ. The data support the conclusions and my comments are minor.

  1. please alter the presentation of the data in Fig. 3F to include the actual data points to help us appreciate the raw data. please explicitly state p values.

The point of the reviewer is valid. This suggestion has been taken well and specific data points along with the respective P values have been added in the figure 3F. Accordingly, legend of Figure 3F is also modified, please see line 430.

  1. what is the predicted protein from the deletion shown in Fig.3C?

The reviewer has raised a very valid question. Actually, deletion of 78 bp (c.3541_3618del) of CENPJ results in in-frame deletion of 26 amino acids, p.(Val1181_Val1206del). This information, p.(Val1181_Val1206del), has been added in the figure 3C, which made the figure more informative and self-explanatory.

  1. I think this is a great opportunity to robustly discuss the need and opportunity to follow up these genetic associations with biological studies to support the bi-alleleic effects. Some of that such work is done here but a call to action for this would be well placed here.

I am grateful to the reviewer to indicate this very important point. The suggested argument has been inserted in the section of discussion. Please follow the lines 508-512. This has improved the manuscript for the readers and continue the functional studies for further validation.

Sentences added at lines 508-512 are: ‘Hence, these data lay a foundation for the future studies to investigate modifying variants in MCPH cases. Importantly, the phenotypic severity corroborated with sub-cellular anomalies confirmed by our experiments models an efficient strategy for future studies for validation of modifying variants at functional level.’

Reviewer 2 Report

Major

  • Materials and Methods section should include subheadings (i.e., Sequencing, Quantitative real-time reverse transcriptase PCR, Western blotting, Immunohistochemistry, Statistics, etc.) for clarity.

General comments

  • Was there a specific density the 12 mm coverslips were seeded at?
  • Table S1 might be important enough to be used in the main text, not as a supplement.
  • Figure 3E: western blotting for CENP-J could be improved – the bands here do not show the typical reduction in immunoreactivity (fainter bands in comparison to control), but more misshapen.

Specifics

  • Line 39: Define abbreviation MCPH.
  • Line 59: Define SD (standard deviation).
  • Figure 1: Why is there a “2” in individual IV-3 in Family 5?
  • Lines 40-41: “It has been hypothesized that additional genetic variants may be responsible for this heterogeneity but data are sparse” this sounds a bit awkward, I suggest changing it to “is sparse”
  • Line 179: can you name the neurodevelopmental disorders affected by these genes of interest?
  • Lines 195-197: “Additional phenotypic presentation of lock joints, drooling and short stature, which were observed only in a few affected members of this family, prompted us to look for additional potentially disease-causing variants” – “prompting us to look”
  • Line 275: don’t spell out the previously defined acronym (MOPDII).
  • Lines 369-370: “number of experiments are 3 where 100 cells/experiment were considered” – sounds awkward, I suggest “three experiments were considered where 100 cells were visible”

Minor

  • Lines 290-300: section talking about PCNT variant – the authors comment about the conservation of mutated residues for many of the other variants but don’t for this one. Is there a reason why?

Grammar

  • Line 324: “…mutant CENP-J at [the] cellular level.”

Author Response

Below, we are copying Reviewers comments/questions individually, followed by our responses (in italics).

Reviewer 2:

Comments to the Author

Major

  • Materials and Methods section should include subheadings (i.e., Sequencing, Quantitative real-time reverse transcriptase PCR, Western blotting, Immunohistochemistry, Statistics, etc.) for clarity.

The suggested change has been made in the section of Materials and Methods by mentioning subheadings to each experimental strategy. This suggestion has made the section of Materials and Methods informative and easier to follow.

General comments

  • Was there a specific density the 12 mm coverslips were seeded at?

The specific density of the cells have mentioned under the subsection 2.7 (Immunocytochemistry) of Materials and Methods. Please follow lines 188-189.

Sentence is modified as: ‘For immunofluorescence, primary fibroblasts were grown on 12 mm coverslips with a maximum confluency of 70 %.’

  • Table S1 might be important enough to be used in the main text, not as a supplement.

Thanks to the reviewers for recognizing the significance of our supplemental data. The suggestion has been taken and Table S1 is now the main part of the manuscript with a title of Table 2. The corresponding reference of Table 2 has also been modified in the main text as needed. 

  • Figure 3E: western blotting for CENP-J could be improved – the bands here do not show the typical reduction in immunoreactivity (fainter bands in comparison to control), but more misshapen.

In this regard, another blot of the same experiment has been added in supplemental data (Figure S4) as a proof of marked reduction in the mutants as compared to the wild type. We believe that the immunoblot shown in Figure 3E has better quality compared to immunoblot shown in Figure S4. Therefore, we did not replace it. But both immunoblots show the marked reduction of CENP-J in both mutants compared to the wild type.

Specifics

  • Line 39: Define abbreviation MCPH.

The abbreviation of MCPH is included in the abstract. Please follow line number 41.

  • Line 59: Define SD (standard deviation).

The abbreviation of SD is included in the Introduction. Please follow line number 62.

  • Figure 1: Why is there a “2” in individual IV-3 in Family 5?

A numerical inside a square in the given pedigree indicates number of individuals (siblings) of same gender and having similar affection status by the set standards. For the information of the reviewer, it (2) shows two normal brothers in the pedigree of family 5 in Figure 1.

  • Lines 40-41: “It has been hypothesized that additional genetic variants may be responsible for this heterogeneity but data are sparse” this sounds a bit awkward, I suggest changing it to “is sparse”

I would, very humbly, disagree to the argument for the fact that ‘are’ is used for the plural form ‘data’. In case of datum, the suggestion of ‘is’ would have been taken.

  • Line 179: can you name the neurodevelopmental disorders affected by these genes of interest?

The reviewer has indicated a missing piece of information. The sentence is modified as suggested by the reviewer. Please follow line number 442-443. Now the text is more clear for the readers to follow the results section.

The sentence is modified as: ‘the known involvement of these genes in the neurodevelopmental disorders which are discussed below in continuum of the respective modifiers.’

  • Lines 195-197: “Additional phenotypic presentation of lock joints, drooling and short stature, which were observed only in a few affected members of this family, prompted us to look for additional potentially disease-causing variants” – “prompting us to look”

I would, very humbly, disagree as the section of result is written in past tense and that is why the third form of prompt is used in the given sentence.

  • Line 275: don’t spell out the previously defined acronym (MOPDII).

As suggested, the correction is made. Due to this suggestion the redundancy in the text is corrected. Please see lines, 138-139, 335-336 and 455-456.

  • Lines 369-370: “number of experiments are 3 where 100 cells/experiment were considered” – sounds awkward, I suggest “three experiments were considered where 100 cells were visible”

The sentence is modified as per suggested. Please follow line number 432-433.

The modified sentence is: ‘three experiments were considered where 100 cells were accounted for each experiment.’

Minor

  • Lines 290-300: section talking about PCNT variant – the authors comment about the conservation of mutated residues for many of the other variants but don’t for this one. Is there a reason why?

Investigation of amino acids conservation is exclusively informative for missense variants, where wild-type residue is substituted with another functional amino acid. However, PCNT harbors nonsense variation resulting in the early truncation of the protein. Therefore, it is not meaningful to describe the conservation status of PCNT variant. 

Grammar

  • Line 324: “…mutant CENP-J at [the] cellular level.”

The article is added in the suggested sentence.